# Questionable research practices in competitive grant funding: A survey

**Stijn Conix** [1,2]*, **Steven De Peuter** [3], **Andreas De Block** [2], **Krist Vaesen** [4]

**1** Center for the Philosophy of Science and Society, UCLouvain, Louvain-la-Neuve, Belgium, **2** Centre for Logic and Philosophy of Science, KU Leuven, Leuven, Belgium, **3** Methods, Individual and Cultural Differences, Affect and Social Behavior (MICAS), KU Leuven, Leuven, Belgium, **4** Philosophy & Ethics, Eindhoven University of Technology, Eindhoven, Netherlands

* stijn.conix@uclouvain.be

**Data Availability Statement:** The data used in this paper has been uploaded to Zenodo. For the main study, the data is at https://zenodo.org/record/7673890 (DOI: 10.5281/zenodo.7673890). For the

## Abstract

There has been a surge of interest in research integrity over the last decade, with a wide range of studies investigating the prevalence of questionable research practices (QRPs). However, nearly all these studies focus on research design, data collection and analysis, and hardly any empirical research has been done on the occurrence of QRPs in the context of research funding. To fill this gap, we conducted a cross-sectional pre-registered survey of applicants, reviewers and panel members from the Research Foundation–Flanders (FWO), one of the main funding agencies in Belgium. We developed a bespoke survey and further refined it through feedback from experienced researchers and a pilot study. We asked how often respondents had engaged in a series of QRPs over the last ten years. A total of 1748 emails were sent, inviting recipients to participate in the survey, complemented by featuring the survey in the FWO newsletter. This resulted in 704 complete responses. Our results indicate that such QRPs are remarkably prevalent. Of the 496 participants who answered both the applicant and reviewer track, more than 60% responded that they engaged regularly in at least one of such practices, and around 40% indicated that they engaged at least occasionally in half of the QRPs queried. Only 12% reported not to have engaged in any of the QRPs. Contrary to our hypotheses, male respondents did not self-report to engage in the QRPs more often than female respondents, nor was there an association between the prevalence of QRPs and self-reported success rate in grant funding. Furthermore, half of the respondents indicated that they doubted the reliability of the grant peer review process more often than not. These results suggest that preventive action is needed, and provide new reasons to reconsider the practice of allocating research money through grant peer review.

## Introduction

In recent decades, research integrity and in particular breaches of research integrity have been the focus of an increasing number of studies. Although the prevalence of outright fraud–fabrication, falsification and plagiarism–appears to be relatively low at 2 to 4%, so-called questionable research practices (QRPs) are disturbingly prevalent at an estimated 34% [1–3]. One study on researchers in the Netherlands found that 50% engaged frequently in at least one

pilot, the data is at https://zenodo.org/record/6945173 (DOI: 10.5281/zenodo.6945173).

**Funding:** The author(s) received no specific funding for this work.

**Competing interests:** The authors have declared that no competing interests exist.

QRP [2]. Initial research into the causes of QRPs focused on the role of the individual researcher, but neither personal factors nor individual differences in (dis)honesty were found to be strong drivers of questionable behaviour [4,5]. In contrast, the institutional context and the research climate in which individual researchers operate have the largest influence on researchers' behaviour [2,6–8]. Research climate factors explained 22% of the variance in perceived frequency of research misbehavior [8] and has been implicated in promoting *responsible* research behaviours [6,9]. Within the broader research ecosystem, funders and publishers exert their influence as well, for the better or for the worse [7,9].

The majority of existing empirical research on questionable research practices addresses issues relating to research design, data collection and publication practices. However, in the light of funders' influential role in shaping the research ecosystem and taking into account the time and effort invested in distributing funding through competitive calls [10–13], it is surprising how little attention has been devoted to research integrity in the context of peer reviewed project funding. Recently, but based mostly on anecdotal evidence, it has been suggested that the very nature of the system of peer reviewed project funding may not only incentivize, but actually *force* researchers to resort to unacceptable practices [14], resulting in the violation of many, if not all, values that are commonly regarded as central to responsible research conduct–accountability, honesty, responsibility, impartiality, and fairness.

Some examples of questionable practices related to peer reviewed project funding [14] are funding being requested on work already (partly) done; double-dipping or submitting the same proposed work to several funders without letting the funders know; and improper use of funds. Peer reviewers, in turn, may not disclose potential conflicts of interest; accept invitations to review proposals outside their area of expertise; invest insufficient time to properly assess a proposal; take ideas from the proposals they review or veto proposals that contest the reviewer's own work. One can argue that many QRPs in the context of funding are situated in a *grey zone* between acceptable and unacceptable. For example, involving junior researchers in the writing of funding proposals can be part of a valuable learning process. However, when the junior researcher is not given due credit, problems with *accountability*, *honesty* and *respect* as they are described in various prominent codes of conduct do arise [14,15]. In this study, we will consider all behaviours that directly violate generally accepted scientific codes of conduct to be QRPs, even if some of these are considered "normal misbehaviours" [16].

As previously mentioned, there is only anecdotal evidence that applicants and reviewers do not always act in accordance with commonly accepted codes of conduct [16–19]. In this study, we aim to quantitatively test this claim. In particular, we use a survey to get a sense of the prevalence of questionable research practices in the context of peer reviewed project funding. To that end, we sent out a bespoke questionnaire through the Research Council–Flanders (FWO) to reviewers, panel members and applicants from all career stages and scientific disciplines, asking them about their own behaviors and experiences in the peer review funding process over the past decade.

## Methods

### Open science

No personal data were collected for the main study, and data collection for the pilot and main study only started after the full research design was preregistered on the Open Science Framework (OSF). We obtained written informed consent through the landing page of the study which, like the invitation email, explained the aims and design of the study. The full questionnaires, analysis plans, analysis code and supplementary materials for both the pilot and main study can be found on the OSF page (https://osf.io/jk6wd/) of the research project. The full datasets of both the pilot and main study can be accessed on Zenodo [20,21].

## The questionnaire

We used an online questionnaire (available in S1 File) to collect self-report data from academic researchers concerning their behavior in the grant peer review process. The questionnaire was developed through an iterative process. Starting from an exhaustive list of questionable research practices (based on [14]), we selected the practices we expected to be the most pressing and common, and had these checked by a copy-editor and experienced researchers for face validity. We then ran a pilot study among researchers who served in recent years as panel member at the European Research Council (N = 139, response rate = 32%; instrument available on the OSF page) before making final changes based on the comments we received on the pilot. Because we do not assume there is one underlying psychological construct the items tap into, we did not perform factor analysis nor did we check internal consistency in other ways.

The questionnaire consisted of three separate blocks of questions aimed at reviewers, applicants and panel members, respectively. To check for eligibility, respondents had to indicate the roles they had taken up in the past 10 years, and were then presented with only the blocks corresponding to those indicated role(s).

In each of the three blocks, we distinguished between questions about QRPs and questions about participants' experiences in the funding system ("information questions" henceforth). Across roles, all QRP questions directly asked how often the respondent had engaged in a certain practice in the past decade (e.g., '*In the past decade, how often have you intentionally cited works of potential referees in order to improve your chances of securing a grant*?'). Response options for these questions ranged from 1 ('never') to 7 ('almost always') without intermediate labels; a NA-option was added where relevant. We used a 10-year period rather than the shorter 3-year period that is often used in similar studies [2,3] because researchers presumably spend more time on research than on writing grants, and hence are more often in a position to engage in QRPs in their day-to-day research practice.

"Information questions" were questions about the respondents' general experiences in the funding system (e.g., *In the past decade, how often have you received a grant-review report which you thought accurately reflected the quality of your grant proposal*?). The response options for most of these information questions were the same as those for QRP questions (viz., a 1–7 scale). However, some of the information questions required different scales (e.g., *On average, how confident are you about the reliability of your evaluations of grant proposals*?).

In addition to information questions and QRP questions, the survey contained eight questions about respondent demographics that, on the basis of existing literature, we expected to be potentially associated with the prevalence of QRPs: gender, field of research, location of professional activity, seniority, success rate in funding over the last decade, number of applications submitted over the last decade, and the extent to which the respondent, at some point over the last decade, had had too little funding to properly do research. For each of these questions, the response options were intentionally vague (e.g., for "location of professional activity", we used continents rather than countries) to guarantee anonymity for the respondents. While additional detail may have been desirable, we prioritized anonymity because of the sensitivity of the topic of the questionnaire. For the same reason, respondents could skip each of the questions without abandoning the survey.

## Data collection

The survey was implemented in Qualtrics (version: June 2022). The weblink to the survey was distributed by FWO, the main funding agency in the Dutch speaking northern region of Belgium. In 2021, FWO received 1297 project applications and eventually awarded €120 million to 266 fundamental research projects [22]. FWO sent out the survey using an invitation email

(template available in S2 File) containing the research aims, our contact details, and a link to the landing page of the survey that sought participants' informed consent. In addition, the email incentivized participation by promising a donation on our part of one euro per completed response to an effective charity (total donation of €917 to the Malaria Consortium).

FWO distributed this email over three channels. First, to their 748 panel members for junior and senior research projects, on 29/09/2022 with a follow-up reminder two weeks later. Second, via their partner Science Direct, which FWO uses to recruit reviewers, to a subset of 1000 randomly selected reviewers of FWO projects on 10/10/2022 with a reminder two weeks later. Finally, a link to the survey was posted in the FWO monthly newsletter, which was published on the FWO website and also sent out to all applicants on September 29, and repeated one month later on October 27 (number of recipients unknown). The survey was closed on 10/11/2022, two weeks after the final newsletter call.

Respondents who took less than one minute to complete the entire questionnaire or who did not complete at least half of the QRP questions of one or more roles were removed from the dataset. For all hypothesis tests and regression analyses, responses with missing data for any of the included questions were removed from the dataset. The data from the pilot study (n = 139), which had a slightly different set of questions, were not included in the analysis for this study.

The data were downloaded through QualtricsAPI [23], a python wrapper for the Qualtrics API, directly into a Jupyter notebook for analysis.

## Outcome variables and aggregate measures

To investigate associations between respondent characteristics and the prevalence of QRPs, we used four outcome variables, namely, two respondent-level and two item-level ones. Although utilizing four different outcome variables increased the level of complexity of the analysis, we deemed it crucial to encompass different aspects of the prevalence of QRPs.

The two respondent-level outcome variables aggregate all QRP responses across the reviewer and applicant blocks for respondents who completed both these blocks and the respondent characteristic questions about field, gender and seniority. We did not analyze the QRP questions separately for applicants and reviewers because there was no clear thematic similarity between the QRPs associated to each of these roles. Lacking such thematic similarity within roles, we considered it preferable to merge QRPs across roles in the outcome variables in order to get a view on the participants' overall tendency to engage in QRPs. We did not add the panel member QRP to the aggregate measures because we expected fewer people to complete the panel member question block than the two other blocks.

The first respondent-level outcome variable was a binary measure tracking whether participants scored at least '4' (out of 7) for at least one QRP ("FREQ" henceforth). This FREQ indicator tracks whether there are QRPs that the participants engage in on a regular basis. The second respondent-level outcome variable was a binary measure tracking whether respondents scored at least '2' (out of 7) for at least half of all the QRPs ("HALF" henceforth). This HALF indicator tracks whether the participants occasionally engage in many of the QRPs. We considered both aggregate measures important to get a grasp on two distinct dimensions of the prevalence of QRPs, namely, committing many QRPs occasionally, and committing some QRPs frequently.

The two item-level outcome variables were on the one hand the item's ordinal response options ranging from '1' (never) to '7' (almost always) with 'NA' recoded as 1 (never) and, on the other hand, a binary recoding of each item's response into 'never' (ordinal score of 1 or 'NA') versus 'at least sometimes' (ordinal score of 2 or higher; "ALS" henceforth).

Although the main aims of this study were exploratory, we did preregister two hypotheses: we expected 1) *respondents who self-identify as male report to have committed more QRPs than respondents who self-identify as female* and 2) *researchers with a higher self-reported success rate in securing funding report to have committed more QRPs than researchers with lower success rates.* The first hypothesis is based on previous studies showing that male researchers are more often involved in scientific misconduct and QRPs [2,24] and on the findings of the pilot study we conducted in preparation of the survey reported here. Although the role of gender in research misconduct has been challenged [25,26], we believe the presence of the effect in the large sample of the Gopalakrishna et al. study [2] and in our pilot justify hypothesizing a gender effect. We tested the second hypothesis mainly for conceptual reasons: we expected that securing funding constitutes the main motivation for academics to engage in the QRPs queried here, and that engaging in them may be associated with improved chances of securing funding. There are to date, as far as we know, no published reports describing the relationship between applicants' success rate and QRPs, but in our pilot study we observed a weak increase of QRPs with higher success rates, particularly for researchers with the highest self-reported success rates (i.e., > 75%).

## Analysis

All analyses were done in Jupyter notebooks in Python using Pandas [27], Scipy [28] and Numpy [29], and Seaborn [30] and Matplotlib [31] for plotting results. All hypothesis tests and explorative regressions were done using the Pymc [32], Bambi [33] and Arviz [34] libraries in python (see S3 File for all package versions).

To make sure that the hypothesis tests were robust and not dependent on the particular way QRP scores were aggregated, we tested both hypotheses using multiple outcome variables. To test the hypothesis that women engage in QRPs less frequently than men, we ran Bayesian logistic regressions with FREQ and ALS as the outcome variables, and gender, seniority and field as predictor variables. In addition, we also ran an ordered logistic Bayesian regression with QRP question responses (1–7) as the outcome, and gender, field, seniority, respondents and question types as predictor variables. Thus, the gender hypothesis was tested by means of three subhypotheses in total. Because only a very small proportion of respondents (2.29%) did not identify as either male or female, these respondents did not generate sufficient data for meaningful analyses and were, as preregistered, removed for the hypothesis tests.

To test the hypothesis that respondents who report higher success rates engage in QRPs more frequently than those who report lower success rates, we ran two Bayesian ordered logistic regressions with success rate (ordinal scale of percentage-intervals) as the outcome variable, and gender, seniority, field and FREQ or HALF respectively as predictor variables. Thus, the success hypothesis was tested by means of two subhypotheses.

For all five subhypotheses, we preregistered one main model and several alternative models to check the robustness of the results. The causal assumptions we made for evaluating both the gender and success hypothesis are expressed in directed acyclic graphs (DAGs) in S1 and S2 Figs respectively. These DAGs were made on the basis of existing literature on QRPs, which suggests that field, gender and seniority, unlike other demographic characteristics, may be associated with the prevalence of QRPs in scientific research [1,2]. The variables included in the models were selected based on these DAGs, using the so-called "backdoor criterion" [35]. For both hypotheses, we focus on the direct rather than total effects, as these are more likely to be useful for explaining differences.

We used weakly informative priors in all models, based on the results of the pilot study. All the analyses were accomplished using Markov chain Monte Carlo methods (MCMC; [36]).

**Table 1. Models for testing the gender and success hypotheses.**

| Model | Outcome variable | Predictor variables | Regression | Effect of interest |
|---|---|---|---|---|
| **Gender test 1** | Item QRP score | Gender, field, seniority, respondent, question type | Ordered logistic | Direct effect gender |
| **Gender test 2** | FREQ | Gender, field, seniority | Logistic | Direct effect gender |
| **Gender test 3** | ALS | Gender, field, seniority | Logistic | Direct effect gender |
| **Success test 1** | Success rate | Gender, field, seniority, FREQ | Ordered Logistic | Direct effect FREQ |
| **Success test 2** | Success rate | Gender, field, seniority, HALF | Ordered logistic | Direct effect HALF |

For details about the likelihood functions and parameters for the main model for each of the five subhypotheses, see Table 1. For the detailed specification of all the models, see S3 File and the analysis code linked to the OSF project (https://osf.io/jk6wd/).

In addition to these hypothesis tests, we exploratorily ran various Bayesian regressions with the same outcome variables and different predictor variables. More precisely, we explored the relation between field of research and the prevalence of QRPs, as previous research suggests that QRPs are more prevalent in the biomedical sciences than in other fields [2,26], and the relation between lack of funding and QRPs, as we expect that researchers who lack sufficient funding might engage in QRPs more often than researchers who have sufficient funding.

# Results

## Survey info

A total of 1748 emails was sent out by FWO and Science Direct (748 to FWO panel members, 1000 to FWO reviewers) with an invitation to take part in the survey. In addition, the survey was also included as an item in the FWO monthly newsletter. Neither panel members nor reviewers need to be affiliated to Flemish institutions by the FWO rules, but a substantial proportion of the former typically are. The vast majority of the reviewers are affiliated to non-Flemish institutions. A total of 753 respondents started the survey. After excluding responses with too many missing data, submissions after closure of the survey and responses that took less than a minute, 704 responses were retained (670 reviewers, 541 applicants and 487 panel members). Of these 704 respondents, 496 filled in all the QRP questions for both the reviewer and applicant track as well as the respondent characteristic questions about field, seniority and gender.

Because we cannot ascertain how many people received the newsletter, we cannot reliably estimate the response rate (but see S3 Fig for how responses peak after the different distribution methods). It should be noted, however, that the pilot study (32%) and similar surveys in The Netherlands [2] and Norway [37] had reasonably high response rates.

Despite the sensitive nature of the questions, the option to skip questions and our policy of retaining even participants who did not complete the survey, missing data were very low (below 2%) across all groups of questions (see S4 Fig). Most of these non-responses were from respondents who indicated playing more than one role, and who did not fill out the questionnaire for all their roles. The only question that stood out with a higher proportion of missingness (namely, 4.40%) was the question about the estimated success rate in funding applications.

## Respondent characteristics

As could be expected from a survey distributed by a Belgian funding agency, respondents were mainly professionally active in Europe (77.08%). Other than this, participants were mostly divided between fields and levels of seniority as could be expected on the basis of the

population that received invitations to participate. In particular, the observed proportions of responses from the various fields match closely with the proportions in that population (see S2 Table), and the relatively low proportion of researchers with less than 10 years of seniority should be expected from a population of which around half serves as a panel member. S1 Table fully describes the distribution of field and seniority by gender. S5 Fig fully describes all demographic respondent characteristics. About half (50.14%) of all respondents indicated that, over the past decade, they more often than not lacked sufficient funding to do meaningful research. Compared to the median of 3.5 (out of 7), these numbers were higher for researchers from Life & Biomedical Sciences, researchers identifying as female, and researchers with less than 30 years of seniority since their PhD (all median = 4). Life & Biomedical Sciences also stood out in terms of the number of applications submitted (median of 11–20, compared to 6–10 overall). S6 Fig fully describes the distributions of lack of funding, number of applications and success rates among participants.

## Prevalence of QRPs

Across all QRPs, between 20.21% and 75.23% (mean = 46.48%) of all respondents indicated to engage in the QRP at least sometimes. Between 1.35% and 28.31% (mean = 13.31%) even reported to do this more often than not (a score of at least 4). Among the 496 participants that filled in all applicant and reviewer QRP questions, 67.34% scored 4 for at least one QRP, and 41.73% indicated to engage at least sometimes in at least half of the QRPs. Of all QRP question responses, 45.39% were 2 or higher, indicating to have committed the QRP at least sometimes over the past decade. Only 12 (2.43%) participants indicated not to have engaged in any of the QRPs over the past decade.

The most prevalent QRPs to be committed on a regular basis (a score of at least 4) were overstating confidence in the predictions made in research proposals (28.31%), improper use of funds at the end of a project (21.05%) and putting insufficient effort in reviewing an application (19.16%). The most prevalent QRPs to be committed at least sometimes were overstating confidence in the predictions made in a proposal (75.05%), putting insufficient effort in reviewing an application (70.05%), and not preparing sufficiently for a panel meeting (64.44%). Fig 1 shows the prevalence of all QRPs.

Exploratory analysis of the associations between aggregated QRPs and respondent characteristics suggest that, particularly for FREQ, respondents from Life & Biomedical Sciences were substantially more likely to have engaged in QRPs than respondents from all other fields, and in particular than respondents from Arts & Humanities (see S7 Fig). There was also a weak association between the lack of funding and the tendency to engage in QRPs, although not for those who lacked funding most often (see S8 Fig). Note, however, that these were not preregistered hypotheses, and thus should be taken as potential patterns to be further explored in hypothesis-based research.

## Information questions

In addition to questions about the prevalence of QRPs, respondents reported their experiences with the funding process. Concerning the reliability of peer review, 52.10% of all 704 respondents scored the likelihood of getting a bad reviewer (i.e., not an expert and/or with a conflict of interest) as 4 or higher. In line with this, scores of at least 4 for the prevalence of getting an unfair negative review report (52.46%), non-expert reviewers (61.74%), and inaccurate review reports (59.47%) were relatively common. Similarly, scores of at least 4 for the prevalence of observing ill-prepared panel members (41.60%) and proposals that are impossible to compare (35.43%) were high among panel members. The complete responses for all information

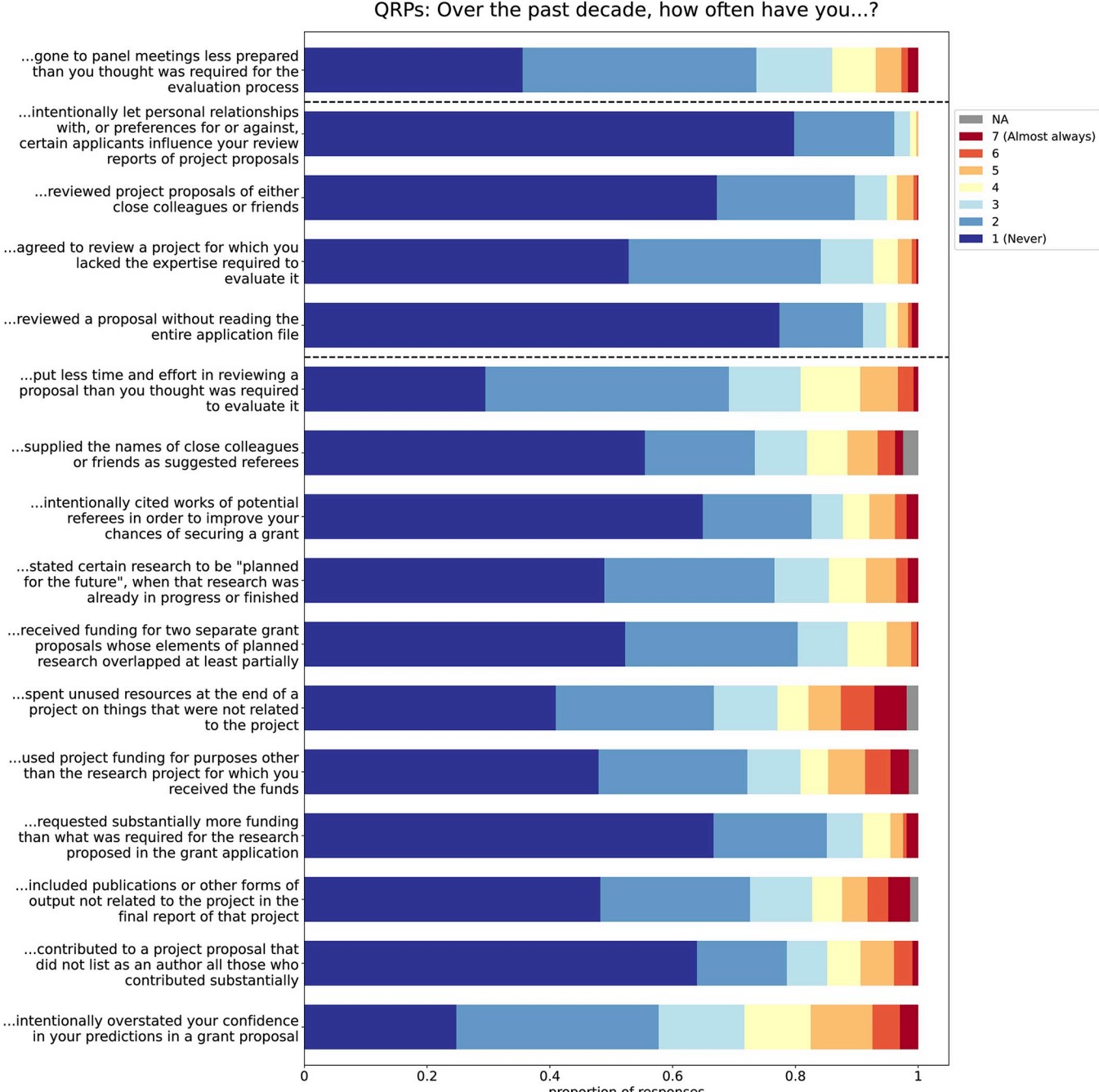

**Fig 1. Responses to all QRP questions.** The questions below the lower dashed line are the applicant QRPs, the questions between the dashed lines are the reviewer QRPs, and the top question is the panel member QRP.

questions that were on a 1–7 scale are summarized in Fig 2. S9 and S10 Figs summarize the responses to the information questions that were not on a 1–7 scale.

## Hypotheses

The potential scale reduction factor (PSRF [38]), also called R-hat, was 1.0 for all parameters in all models, indicating that in each case the three chains converged. The effective number of

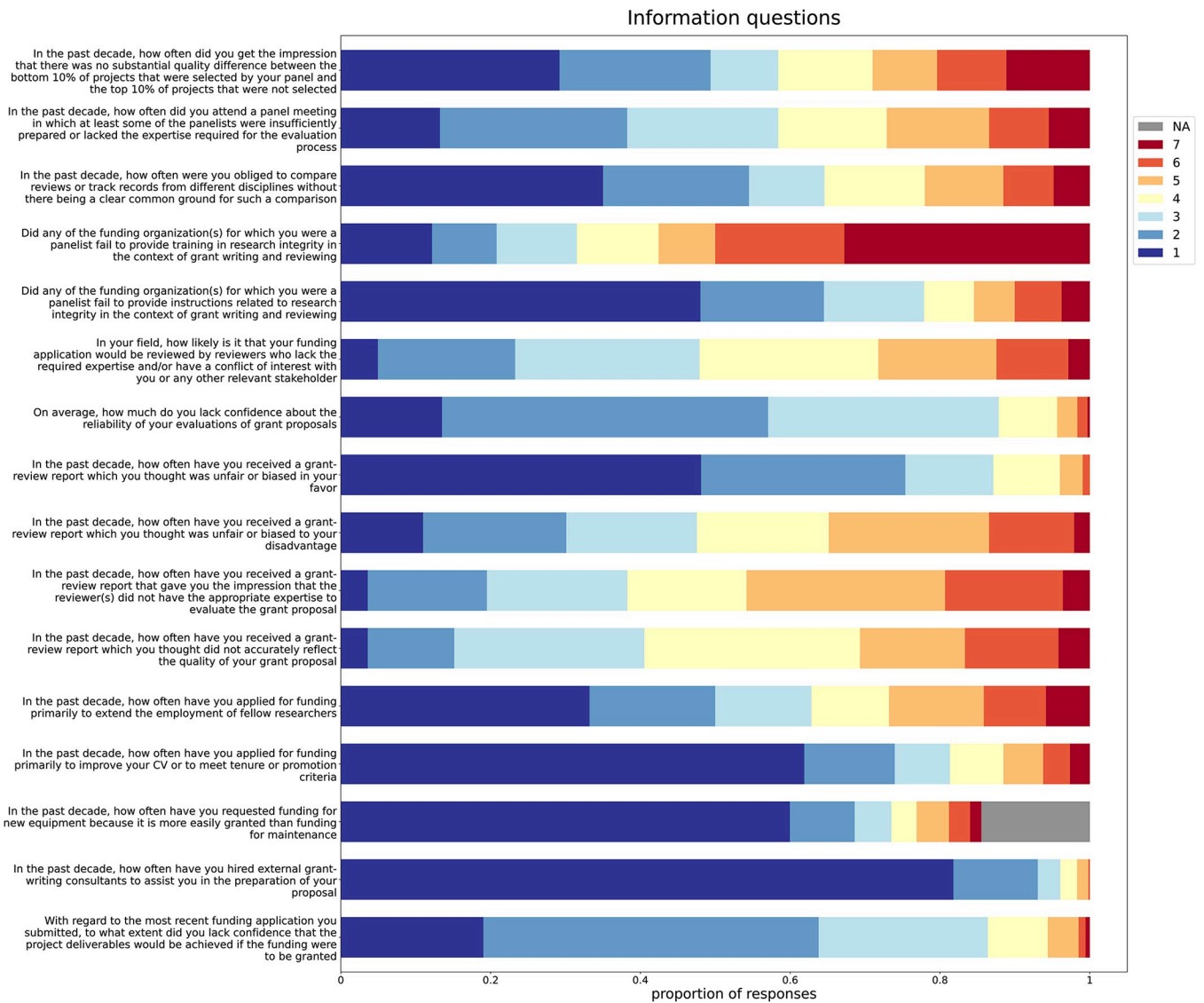

**Fig 2. Responses to all Information questions on the 1–7 scale.**

steps (ESS [39]) in the MCMC chain was above 10,000 for all parameters in all models, suggesting that the estimates are stable and reliable even for the limits of the highest density intervals (hdis). Posterior predictive checks for all models show that the models do well in mimicking the data. For details on the PSRF, ESS and posterior predictive checks, consult the analysis code on the OSF project page, or see S3 File.

The full dataset for the gender hypothesis tests with aggregate outcome variables contained 496 complete responses of all applicant and reviewer QRPs and respondent characteristics from respondents identifying as male or female. The dataset for the gender hypothesis test with the item-level outcome variable contained a total of 8840 responses. Neither of the three hypothesis tests confirmed our hypothesis that men are more likely to engage in QRPs than women. Across all measures respondents identifying as male were even less likely than those identifying as female to report engaging in QRPs. Table 2 summarizes the posterior distributions of the coefficients for the two gender variables (male and female) for the main model of

**Table 2. Regression coefficients for the variables of interest of the hypothesis tests.**

| | Variable | Coefficient Mean | Standard deviation | 5.5% hdi | 94.5% hdi | ESS (bulk) | RHAT |
|---|---|---|---|---|---|---|---|
| **Gender1** (outcome: qrp scores) | Male | -0.644 | 0.237 | -1.047 | -0.186 | 8800 | 1.0 |
| | Female | -0.614 | 0.242 | -1.055 | -0.147 | 8807 | 1.0 |
| | Difference[a] | 0.03 | 0.108 | -0.172 | 0.234 | / | / |
| **Gender2** (outcome: FREQ) | Male | 0.370 | 0.330 | -0.290 | 0.959 | 12049 | 1.0 |
| | Female | 0.764 | 0.339 | 0.104 | 1.387 | 12761 | 1.0 |
| | Difference[a] | 0.394 | 0.204 | -0.001 | 0.767 | / | / |
| **Gender3** (outcome: ALS) | Male | -0.329 | 0.288 | -0.881 | 0.202 | 4452 | 1.0 |
| | Female | -0.262 | 0.292 | -0.810 | 0.286 | 4556 | 1.0 |
| | Difference[a] | 0.067 | 0.108 | -0.134 | 0.271 | / | / |
| **Success4** (outcome: FREQ) | FREQ = 1 | 0.439 | 0.624 | -0.772 | 1.642 | 25538 | 1.0 |
| | FREQ = 0 | 0.644 | 0.644 | -0.649 | 1.764 | 25547 | 1.0 |
| | Difference[a] | -0.205 | 0.177 | -0.534 | 0.127 | / | / |
| **Success5** (outcome: HALF) | HALF = 1 | 0.686 | 0.610 | -0.451 | 1.833 | 19003 | 1.0 |
| | HALF = 0 | 0.810 | 0.609 | -0.346 | 1.940 | 18624 | 1.0 |
| | Difference[a] | -0.124 | 0.164 | -0.432 | 0.183 | / | / |

[a] For the gender hypothesis tests, the difference equals the posterior of 'Female' subtracted by the posterior of 'Male'. For the success hypothesis tests, the difference equals the posterior of 'FREQ'/'HALF' = 1 subtracted by the posterior of FREQ'/'HALF' = 0.

each of the subhypotheses about the association between prevalence of QRPs and gender. Fig 3 shows the difference between the 'female' and 'male' coefficient for each of the three models, as well as the counts of QRP scores from posterior predictive samples with the entire study population changed in turn to 'male' and 'female'. The full results can be found in S3 File, but note that these models were designed to investigate the effect of gender. Hence, coefficients of the other variables are not always directly interpretable [40].

The full dataset for the success hypotheses contained 493 complete responses of all applicant and reviewer QRPs and respondent characteristics. Neither of the two hypothesis tests

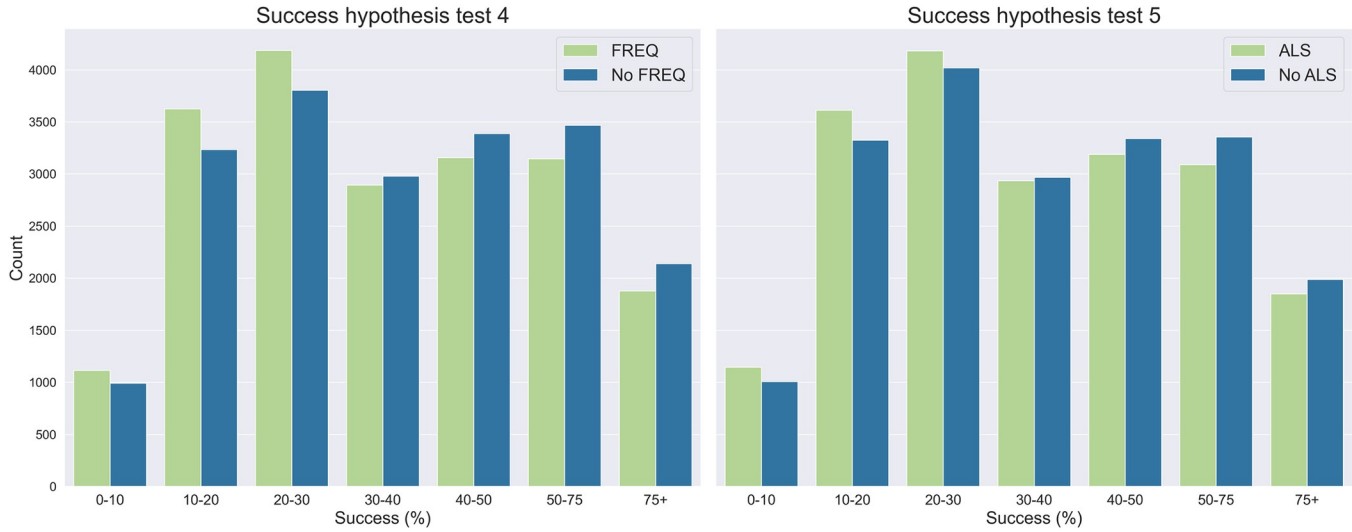

**Fig 3. Posterior predictive counts for the gender hypothesis tests.** These posterior predictive samples were drawn from each of the models, changing our entire population to 'male' and 'female' in turn, keeping the other demographic characteristics intact.

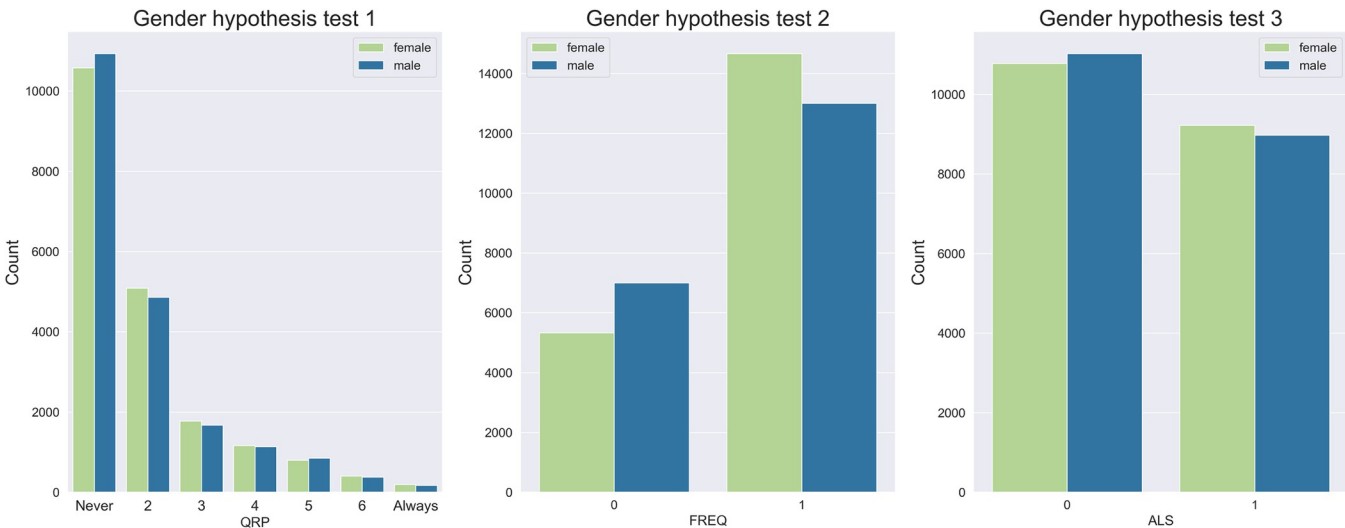

**Fig 4. Posterior predictive counts for the success hypothesis tests.** These posterior predictive samples were drawn from each of the models, setting our entire population to 'FREQ/ALS' = 1 and 'FREQ/ALS' = 0 in turn, keeping the other demographic characteristics intact.

confirmed our hypothesis that the prevalence of QRPs increases with higher self-reported success rates in grant funding. That is, respondents reporting higher success rates were not more likely than those reporting lower success rates to have at least one QRP with a score of 4 (FREQ) or half of all QRPs with a score of at least 2 (HALF). Table 2 summarizes the posterior distributions of the coefficients of the FREQ and HALF parameters. Fig 4 shows the difference between the 'QRP = 1' and 'QRP = 0' coefficient for both models, as well as the counts of QRP scores from posterior predictive samples with the entire study population set in turn to 'QRP = 1' and 'QRP = 0'. The full results can be found in S3 File.

## Discussion

Our survey results suggest that QRPs are widespread in grant writing and grant peer review. For example, of the 496 participants who completed all applicant and reviewer QRP questions, more than two thirds indicated that they regularly engage in at least one QRP. Additionally, more than 40% of these participants reported to engage at least occasionally in half of all QRPs. Moreover, many of the surveyed QRPs had a high number of researchers who reported to engage regularly in them.

Given the limitations of the sample (see below) and the purely descriptive setup of this study, our hypothesis tests do not warrant causal claims about what drives QRPs. Still, our tests indicate that previous hypotheses about the association between gender and QRPs [2,24] do not hold, at least in our sample, in the context of grant peer review. We also did not find the hypothesized association between funding success and QRPs. Exploratory analysis does show some association between field of research and QRPs. Scholars from Life & Biomedical Sciences in particular seem to score higher on QRPs than scholars from other disciplines. This is in line with previous research on QRPs in other parts of the research process, which also suggests that the Biomedical Sciences had the highest frequency of QRPs [2,26]. As researchers in the Life & Biomedical Sciences also scored highly on lack of funding and number of applications submitted, there may be a relation between (hyper-)competition and QRPs. However, further research is required to verify our findings and to gain an understanding of the real drivers of the relationships between research domains and QRPs in the context of peer reviewed project funding.

There are two important take-aways from this study. First, the prevalence of QRPs in peer reviewed project funding is high and problematic. Hence, creating awareness and conversation about these practices [41], and explicitly asking researchers to report (e.g., in funding applications) their (refraining from) engagement in QRPs can already be an important step in reducing their prevalence. Training reviewers and panel members on QRPs in project funding might also be advisable [42]. Since competition seems to be an important driver of QRPs [2], reducing competition may also help to decrease these QRPs. For instance, funding agencies can contribute to a less competitive research environment by striving for a more equal distribution of the available funding [12,43]. Currently, some funding agencies already have explicit regulations regarding the QRPs that our survey queried. For instance, the FWO already explicitly asks applicants to report whether they have submitted the current or a similar proposal elsewhere. Moreover, FWO's panel members are required to state beforehand that they will refrain from sharing the information from grant applications (e.g., novel research ideas or methodologies), and refrain from utilizing this information for personal purposes. Of course, many funding agencies could do more to control the different stakeholders (applicants, awardees, reviewers, etc.) but the moral and financial costs of such more extensive policing might outweigh the benefits [44].

Maybe the most efficient countermeasures can be taken by research institutions, as they are usually the employers of grant applicants. Currently, they often increase the already high stakes of the funding game, for example by making acquired funding an important factor in tenure decisions or salary negotiations [14]. This contributes to a very competitive research environment that is conducive to QRPs. Indeed, institutions may even sometimes inadvertently encourage QRPs, for example (and anecdotally) when PIs are put under pressure to spend 'unused' funds, so that these do not flow back to the funder. Probably, part of such institutional misbehavior can be explained by the fact that many of the discussed QRPs appear to be in a moral gray zone. These misbehaviors may even be perceived as acceptable because they are the descriptive norm [45], as is also suggested by some of our results. Still, even normal or normalized misbehaviors remain problematic, and should at least be critically assessed by academics and academic institutions [16].

A second important take-away is that many researchers express serious doubts about the functioning of the system of peer reviewed project funding. The answers to the information questions show that researchers tend to have little confidence in peer review, often receive what they perceive as low-quality reviews, and fairly often fail to invest sufficient effort when reviewing proposals. Relatedly, researchers often doubt the accuracy of reviews, panelists admit that the projects that they have to rank are very often hard to compare, and reviewers acknowledge that they sometimes lack the expertise to review the project proposals that they review. This is striking because researchers are likely to overestimate their effort and ability to accurately review projects [46]. When researchers express doubts about these aspects, it indicates a potentially larger problem. It is also noteworthy that these findings are in line with existing evidence that there is little or no relation between the ranking of projects by peer review and long-term scientific success [47–50], that review scores differ substantially between reviewers [51,52], and that acquiring funding highly depends on which particular reviewers do the review [53,54].

As half of the surveyed researchers also stated that, more often than not, they generally lacked sufficient funding to do meaningful research, these results might even suggest that more drastic changes to the system of peer reviewed project funding are required. It has already been noted that issues such as the high opportunity costs and lack of reliability of peer review are largely absent from proposed alternatives such as lottery or baseline funding [12,13,43,55]. Importantly, these alternative funding mechanisms are also not plagued by

many of the reported QRPs that seem to be endemic to peer reviewed project function. Lottery and baseline funding systems do not let researchers overstate their confidence in their research, do not produce unfair review reports, and do not require reviewers or panel members to compare proposals that are impossible to compare. Moreover, these alternatives create a less competitive research climate, and may thus be preferable over systems that incentivize the surveyed QRPs. On the other hand, lottery and baseline funding may not always allocate funding optimally, a problem these alternatives share with peer reviewed project funding [13].

Note that the results of this study should be interpreted with caution due to several limitations. The most salient limitation is sample representativeness. Even though the sample seemed to represent the various fields of research appropriately, most respondents came from Europe. Additionally, the survey was conducted exclusively with applicants, reviewers and panel members of the FWO funding agency. Although the FWO procedures and success rates may not appear to be out of the ordinary, and most respondents likely also had experience in other funding schemes, this may mean that the results may not be applicable to other funding agencies with different reviewing procedures, success rates, or operating in a different research context. Finally, this is a convenience sample. While we have no theoretical reasons to assume a selection bias in the sample, this cannot be excluded.

Another limitation is that this study might well underestimate the prevalence of QRPs. It is known that even in anonymous surveys, participants do not always respond honestly due to social desirability bias or fear of consequences [56]. We primed the participants to regard the practices we queried them about as *questionable* research practices, so that, given social desirability and fear of consequences, participants might have understated the extent to which they engaged in these practices. On the other hand, it is widely known that the consequences of, for instance, overstating confidence or improper use of research funds are minor or even non-existent, and that important stakeholders, such as in-house funding advisors/consultants in this process actually and knowingly incentivize some of these QRPs (such as double-dipping). In addition, one study even found that QRPs deemed necessary for career success tended to be seen as not very unethical [57]. This may entail that not all QRPs discussed here are taboo, and some of them are probably not even seen as QRPs [58].

## Supporting information

**S1 Fig. Gender hypothesis DAG.** Directed acyclic graph that expresses our causal assumptions for the gender hypothesis tests. For the hypothesis tests with aggregated variables, the 'Respondent' and 'Question' variables and the edges connected to them should be removed.
(TIF)

**S2 Fig. Success hypothesis DAG.** Directed acyclic graph that expresses our causal assumptions for the success hypothesis tests.
(TIF)

**S3 Fig. Timeline survey responses.**
(TIF)

**S4 Fig. Missing data by question type.** Proportion of respondents that were asked the question but did not answer it. Hence, this does not include questions for roles that the respondents did not indicated they had played.
(TIF)

**S5 Fig. Demographic characteristics of the sample.**
(TIF)

**S6 Fig. Funding characteristics of the sample.**
(TIF)

**S7 Fig. Differences between fields in QRPs.** The top row shows the posterior distribution of the differences in coefficients between 'Life and Biomedical Sciences' and the other fields. The bottom row shows the distribution of QRP item response scores for 20000 posterior predictive samples drawn from the study population but setting their field in turn to each of the fields.
(TIF)

**S8 Fig. Differences between levels of lack of funding in QRPs.** The two top row shows the posterior distribution of the differences in coefficients between 'No lack of funding' and the other levels. The bottom row shows the distribution of QRP item response scores for 20000 posterior predictive samples drawn from the study population but setting their field in turn to each of the levels of lack of funding.
(TIF)

**S9 Fig. Observing and reporting QRPs.**
(TIF)

**S10 Fig. Dealing with uncertainty in reviews.**
(TIF)

**S1 Table. Seniority and field by gender of the respondents.**
(DOCX)

**S2 Table. Expected and observed proportion of responses by field.**
(DOCX)

**S1 File. Full survey instrument.** The questionnaire exported from Qualtrics into a.docx file.
(DOCX)

**S2 File. Invitation email.** Email sent out by FWO to invite researchers to participate in the survey.
(DOCX)

**S3 File. Html exports of the notebooks with analysis code.** The code can also be accessed through the OSF page of the project (https://osf.io/jk6wd/).
(ZIP)

## Acknowledgments

We are grateful to FWO and, in particular, Frederik Van Acker, for helping us distribute the survey among their panel members, reviewers and applicants. We are also grateful to Lin Li and Gert Storms for insightful comments on a draft of this paper. The data for this paper were generated using Qualtrics software, Version June 2022 of Qualtrics. Copyright © 2020 Qualtrics. Qualtrics and all other Qualtrics product or service names are registered trademarks or trademarks of Qualtrics, Provo, UT, USA. https://www.qualtrics.com.

## Author Contributions

**Conceptualization:** Stijn Conix, Steven De Peuter, Andreas De Block, Krist Vaesen.

**Data curation:** Stijn Conix.

**Formal analysis:** Stijn Conix.

**Investigation:** Stijn Conix, Steven De Peuter, Andreas De Block, Krist Vaesen.

**Methodology:** Stijn Conix, Steven De Peuter, Andreas De Block, Krist Vaesen.

**Project administration:** Steven De Peuter, Andreas De Block.

**Resources:** Andreas De Block.

**Visualization:** Stijn Conix.

**Writing – original draft:** Stijn Conix, Steven De Peuter, Andreas De Block, Krist Vaesen.

**Writing – review & editing:** Stijn Conix, Steven De Peuter, Andreas De Block, Krist Vaesen.

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
