## [Decision Letter · Decision Letter 0]

12 Jul 2023

PONE-D-23-16082Questionable research practices in competitive grant funding: a surveyPLOS ONE

Dear Dr. Conix,

Thank you for submitting your manuscript to PLOS ONE. After careful consideration, we feel that it has merit but does not fully meet PLOS ONE’s publication criteria as it currently stands. Therefore, we invite you to submit a revised version of the manuscript that addresses the points raised during the review process.

We look forward to receiving your revised manuscript.

Kind regards,

Sonia Vasconcelos, PhD

Academic Editor

PLOS ONE

Journal Requirements:

Academic Editor's comments: 

In your revision, please give special attention to the concerns raised by #Reviewer 1 and consider the following additional points:

Clarify survey instrument validation and eligibility criteria. On sample size, avoid guessing or speculating on response rates if you do not have the specific data. Although mentioning response peaks can give readers an idea of survey interest and accessibility, it is important to note that they cannot substantiate the claims made about response rates, especially when you speculate that they are higher than those of two similar surveys. Also, revise your abstract to ensure that the percentages make sense to readers and they can understand the reported data properly.

On the survey instrument and your interpretation of results, consider that some of the issues you explored and then interpreted as questionable research practices might be perceived as normal misbehaviors in science. It does not mean that we should normalize unethical behavior but that some perceptions of questionable research practices cannot be taken for granted (please note the comment of Reviewer #3 on their view on one of the behaviors you categorize as questionable). I recommend that you go through “Normal Misbehavior…” by De Vries et al (2006). 10.1525/jer.2006.1.1.43

With uncertainties inherent to grant review, your observation that “[p]articularly remarkable is that researchers have serious doubts about the accuracy of reviews, panelists admit that the projects that they have to rank are very often hard to compare...” seems to be naive. A golden rule for the assessment of grant proposals will not prevent this kind of problem. I invite you to reflect on assumptions – for example, when you write that the “survey results suggest that QRPs are widespread in grant writing and grant peer review….” note that your own assumptions of QRPs might unintentionally be inflating your results and conclusions. 

Please be mindful of not placing too much weight on anecdotal evidence when supporting your claims. While the anecdotal evidence you describe may provide valuable insights, it is important to consider its limitations and potential biases - those of the authors' included.

Reviewers' comments:

Reviewer's Responses to Questions

**Comments to the Author**

1. Is the manuscript technically sound, and do the data support the conclusions?

Reviewer #1: Partly

Reviewer #2: Partly

2. Has the statistical analysis been performed appropriately and rigorously? 

Reviewer #1: Yes

Reviewer #2: Yes

3. Have the authors made all data underlying the findings in their manuscript fully available?

Reviewer #1: Yes

Reviewer #2: No

4. Is the manuscript presented in an intelligible fashion and written in standard English?

Reviewer #1: No

Reviewer #2: Yes

5. Review Comments to the Author

Reviewer #1: This manuscript reports the findings of an online survey on questionable research practices (QRP) among potential reviewers related to the Research Foundation - Flanders (FWO) in Belgium. A total of 496 respondents completed all the information for analyses. The topic is within the journal scope and requirements, and the manuscript findings add relevant information to the literature. However, the text is not concise and clear, and the manuscript needs major revisions throughout the text, including the Abstract, Introduction, Methods, Results, and Discussion. Some specific comments are listed below.

1) Abstract: Please include more information on on sample size and methodological procedures, as well as the main findings of your data analysis. The conclusion should be more conservative and directly related to the results.

2) Introduction: Please revise the text to justify yours statements with statistics in the first paragraph. In the third paragraph, please include references to support your point. The last two sentences are somewhat overlapping with methodological information and should be rephrased. Some information can be inserted in the next section and vice-versa (some information in the Methods section would be better placed in the Introduction).

3) Methods: Please see comments above and add more information on the methodological procedures: place, sample sizes (for example: how many FWO members are active and available in the newsletter database?). Were the data from the respondents of the pilot study discarded for the final survey?

4) Results: please revise the text to avoid duplication with the tables and highlight the main information.

5) Discussion: This section is very short in relation to the results. Please discuss further the main findings and concerns in relation to the most sensitive items of the survey. Which items are critical? It would be important to have a categorisation of the items and a referenced discussion about possible ways to tackle the problems.

6) Please add a last paragraph to summarise your study conclusions.

7) Some references should be replaced by more recent ones.

Reviewer #2: This work is very interesting and presents a high degree of subjectivity that I believe has not been fully resolved.

The response rate was too low to resolve so many variables. Another situation that was not well resolved was if the questionnaire was sent to a European agency, obviously most respondents would be from the European continent (77.08%). It would be important to know which countries are on each continent, as there are important cultural differences that could be discussed and highlighted.

Another statement that stands out is the one described in line 63 of the introduction. The fact that a junior researcher writes a funding project is part of a learning process, the important thing is that it is accompanied, guided, approved and sent by a senior, so as it is, I do not see it as bad conduct.

6. PLOS authors have the option to publish the peer review history of their article (what does this mean?). If published, this will include your full peer review and any attached files.

Reviewer #1: No

Reviewer #2: No

---

## [Author Response · Author response to Decision Letter 0]

25 Aug 2023

NOTE: this is the same response as in the uploaded word document, and we recommend using that file as it is easier to read than this textbox.

1) Editor's comments: 

We’d like to thank the editor for their helpful comments, which improved the paper. We respond to these comments point by point here:

In your revision, please give special attention to the concerns raised by #Reviewer 1 and consider the following additional points:

• Clarify survey instrument validation and eligibility criteria. 

Response: 

- We added additional information about survey validation in the ‘questionnaire’ section of the methods. The revised text now reads: 

“Starting from an exhaustive list of questionable research practices (based on [14]), we selected the practices we expected to be the most pressing and common, and had these checked by a copy-editor and experienced researchers for face validity. We then ran a pilot study among researchers who served in recent years as panel member at the European Research Council (N = 139, response rate = 32%; instrument available on the OSF page) before making final changes based on the comments we received on the pilot. Because we do not assume there is one underlying psychological construct the items tap into, we did not perform factor analysis nor did we check internal consistency in other ways..”

- The section ‘Data collection’ describes the eligibility criteria in the second and third paragraph (to be an applicant, reviewer or panellist for in the system of peer reviewed project funding; data to be removed in case the respondent did not fill in at least half of the QRP questions of one of these roles). We have made this more explicit now changing the second paragraph of the ‘questionnaire’ section. It now reads: 

“To check for eligibility, respondents had to indicate the roles they had taken up in the past 10 years, and were then presented with only the blocks corresponding to those indicated role(s).”

• On sample size, avoid guessing or speculating on response rates if you do not have the specific data. Although mentioning response peaks can give readers an idea of survey interest and accessibility, it is important to note that they cannot substantiate the claims made about response rates, especially when you speculate that they are higher than those of two similar surveys. 

We have removed the part that speculates about the response rate, and now simply mention the peaks and point out that similar surveys and the pilot had reasonably high response rates.

• Also, revise your abstract to ensure that the percentages make sense to readers and they can understand the reported data properly.

We added a sentence in the abstract about how respondents were invited into the survey and about the number of respondents. It now reads:

" A total of 1748 emails were sent, inviting recipients to participate in the survey, complemented by featuring the survey in the FWO newsletter. This resulted in 704 complete responses. Our results indicate that such QRPs are remarkably prevalent. Of the 496 participants who answered both the applicant and reviewer track, more than 60%..."

• On the survey instrument and your interpretation of results, consider that some of the issues you explored and then interpreted as questionable research practices might be perceived as normal misbehaviors in science. It does not mean that we should normalize unethical behavior but that some perceptions of questionable research practices cannot be taken for granted (please note the comment of Reviewer #3 on their view on one of the behaviors you categorize as questionable). I recommend that you go through “Normal Misbehavior…” by De Vries et al (2006). 10.1525/jer.2006.1.1.43

Response: Thank you for this point, and for the interesting reference! We agree that many of the behaviours we discuss are probably not considered QRPs by many researchers. However, they do directly violate the main codes of conduct, as described in the Conix et al. 2021 (ref 13) paper that we cite. Hence, we think there are strong reasons to consider them QRPs. To make it clearer that these are grey zone cases, and, as De Vries et al. (now cited in the introduction) describe them, ‘normal misbehaviours’, we have added an example, an explicit acknowledgment of this point, and a clear and explicit statement that in this paper we will consider as QRPs anything that violates the principles described by all prominent scientific codes of conduct. It now reads: 

“One can argue that many QRPs in the context of funding are situated in a grey zone between acceptable and unacceptable. For example, involving junior researchers in the writing of funding proposals can be part of a valuable learning process. However, when the junior researcher is not given due credit, problems with accountability, honesty and respect as they are described in various prominent codes of conduct do arise [14,15]. In this study, we will consider all behaviours that directly go against generally accepted scientific codes of conduct to be QRPs, even if some of these are considered “normal misbehaviours” [16].”

• With uncertainties inherent to grant review, your observation that “[p]articularly remarkable is that researchers have serious doubts about the accuracy of reviews, panelists admit that the projects that they have to rank are very often hard to compare...” seems to be naive. A golden rule for the assessment of grant proposals will not prevent this kind of problem. I invite you to reflect on assumptions – for example, when you write that the “survey results suggest that QRPs are widespread in grant writing and grant peer review….” note that your own assumptions of QRPs might unintentionally be inflating your results and conclusions. 

The questions discussed in this particular paragraph are not part of the QRP questions, so they had no influence on the results about the prevalence of QRPs. Hence, we could not explicitly state here that our conclusions about QRPs were influenced by our interpretation of these reliability-problems. However, we understand the more general point and hope that by adding the explicit statement about what we consider QRPs (in the introduction, see previous comment), it is now clearer to the reader what our assumptions are. We have also edited that paragraph (page 18) to soften the phrasing, but have retained the point that our results raise worries about the reliability and accuracy of peer review of research proposals (like other (referenced) lines of research do). 

• Please be mindful of not placing too much weight on anecdotal evidence when supporting your claims. While the anecdotal evidence you describe may provide valuable insights, it is important to consider its limitations and potential biases - those of the authors' included.

There are two places where we mention anecdotal evidence. First, to give an overview of the practices that are mentioned in the literature, and second to give an example. We think that in both cases these are valuable, and would prefer to keep them. We have changed the phrasing, however, to emphasize more strongly that they are anecdotal and not the basis for any conclusions:

- Recently, but based only on anecdotal evidence, it has been suggested that the very nature of the system of peer reviewed project funding may not only incentivize, but actually force researchers to resort to unacceptable practices [14]

- Indeed, institutions may even sometimes inadvertently encourage QRPs, for example (and anecdotally) when PIs are put under pressure to spend ‘unused’ funds, so that these do not flow back to the funder.

2) Reviewer's Responses to Questions

1. Is the manuscript technically sound, and do the data support the conclusions?

Reviewer #1: Partly

Reviewer #2: Partly

2. Has the statistical analysis been performed appropriately and rigorously? 

Reviewer #1: Yes

Reviewer #2: Yes

3. Have the authors made all data underlying the findings in their manuscript fully available?

Reviewer #1: Yes

Reviewer #2: No 

Response: We are not sure which data we have not made available, as all analysis code, data from the pilot, data from the main study, the analysis plans and survey instruments are fully accessible. We would of course be happy to publish anything that is still missing but are not sure what that might be.

4. Is the manuscript presented in an intelligible fashion and written in standard English?

Reviewer #1: No

Reviewer #2: Yes

Response: We’ve had the revised manuscript edited by a professional, native speaking copy editor, and hope that the language problems that reviewer 1 had are now resolved.

3) Reviewer 1

We thank reviewer 1 for their comments and taking the time to review the paper. We respond to their comments point-by-point below:

Reviewer #1: This manuscript reports the findings of an online survey on questionable research practices (QRP) among potential reviewers related to the Research Foundation - Flanders (FWO) in Belgium. A total of 496 respondents completed all the information for analyses. The topic is within the journal scope and requirements, and the manuscript findings add relevant information to the literature. However, the text is not concise and clear, and the manuscript needs major revisions throughout the text, including the Abstract, Introduction, Methods, Results, and Discussion. Some specific comments are listed below.

• Abstract: Please include more information on sample size and methodological procedures, as well as the main findings of your data analysis. The conclusion should be more conservative and directly related to the results.

We have added details about the sampling procedure, sample size and main results (hypothesis tests, prevalence of QRPs and perceived reliability of peer review) to the revised abstract (cf. also response to the editor’s comments). Given the fact that we included more detail about the results in the abstract, we deem the conclusion is warranted and suggest not to alter the final sentence of the abstract.

• Introduction: Please revise the text to justify yours statements with statistics in the first paragraph. 

- We added statistics about FFP and QRPs to the second sentence of the first paragraph: 

"Although the prevalence of outright fraud – fabrication, falsification and plagiarism – appears to be relatively low at 2 to 4%, so-called questionable research practices (QRPs) are disturbingly prevalent with estimates ranging to 34% [1–3] and even more than 50% reporting to frequently engage in at least one QRP [2]."

Furthermore, statistics substantiating the role of the research climate has been added to the second-to-last sentence:

"Research climate factors explained 22% of the variance in perceived frequency of research misbehavior [8] and has been implicated in promoting responsible research behaviours [9,10]".

In the third paragraph, please include references to support your point.

- As mentioned in the second paragraph, "it is surprising how little attention has been devoted to research integrity in the context of peer reviewed project funding". The anecdotal (and some proper empirical evidence) has been summarized in ref [14], and the examples mentioned in the third paragraph come from this paper. We have now added the reference in the first sentence of the third paragraph to make this clearer.

The last two sentences are somewhat overlapping with methodological information and should be rephrased. Some information can be inserted in the next section and vice-versa (some information in the Methods section would be better placed in the Introduction).

We have moved parts from the introduction to the methods section, and hope this makes the introduction clearer. More precisely, we moved the description of the hypotheses to the “outcome variables and aggregate measures” section, where it now directly precedes the statistical analysis.

However, we could not find any parts of the methods section that would be better placed in the introduction, and hence have not moved anything up to the introduction. We would be grateful if the reviewer could clarify what they think should be moved.

• Methods: Please see comments above and add more information on the methodological procedures: place, sample sizes (for example: how many FWO members are active and available in the newsletter database?). Were the data from the respondents of the pilot study discarded for the final survey?

- The section ‘’Data collection” states that “The weblink to the survey was distributed by FWO, the main funding agency in the Dutch speaking northern region of Belgium.” We are not sure which information the reviewer would like us to add here, and are afraid that any further information about the place would be distracting.

- FWO does not know how many email addresses in their database are active, as they include all recent applicants (including those that moved institutions etc.). In addition, the newsletter is automatically posted on their online newsletter archive, so that it is very hard to estimate how many people received the newsletter in the end. We have edited the line about the newsletter to make this clearer: “Finally, a link to the survey was posted in the FWO monthly newsletter, which was published on the FWO website and also sent out to all applicants on September 29, and repeated one month later on October 27 (number of recipients unknown).”

- Thanks for pointing out that it was not clear whether the data from the pilot was included in the results. We now explicitly state in the data collection section that it wasn’t: “The data from the pilot study (n = 139), which had a slightly different set of questions, were not included in the analysis for this study.”

• Results: please revise the text to avoid duplication with the tables and highlight the main information.

We acknowledge that some of the text in the results duplicates information from the tables. However, we do this specially to highlight the main findings, and think this is very common across science. To make this less of a problem, we have moved the table with the seniority and field by gender of the respondents to the supplementary materials (S1 Table).

• Discussion: This section is very short in relation to the results. Please discuss further the main findings and concerns in relation to the most sensitive items of the survey. Which items are critical? It would be important to have a categorisation of the items and a referenced discussion about possible ways to tackle the problems.

We have made small adjustments to the concluding section, but have refrained from adding comments on which items are critical or potential categorizations of the items. The initial design of the study included such a categorization in ‘grave’ and ‘minor’ QRPs, but we quickly realized that classifying the items as such would be very subjective. Because reviewer 3 worries that the paper already is too subjective, we thought it better not to make judgements about which items are more of a problem.

We have edited and extended the discussion of potential solutions as suggested by the reviewer, but have again refrained from evaluating them because this is a very controversial topic that requires more research for very strong claims or definitive decisions. Further opinionizing on the topic might easily be seen as to subjective, as reviewer 2 also suggests. Hence, we thought it better to mention the point, point to the appropriate references, but not elaborate further.

• Please add a last paragraph to summarise your study conclusions.

We have tried such a paragraph, but upon rereading it simply repeated many of the points we had just discussed in that same section. For the sake of space and avoiding repetition, we would prefer not to add such a paragraph. For reference, here is the paragraph; if the reviewer still thinks it is better to add this, we will of course reconsider:

“To summarize, our findings reveal a significant prevalence of QRPs. Contrary to our predictions, there was no notable gender-based disparity in self-reported QRP engagement, and the frequency of QRPs did not correlate with reported success rates in securing grant funding. Additionally, approximately half of the participants expressed recurring doubts about the reliability of the grant peer review process. These outcomes underscore the necessity for pre-emptive measures and provide fresh incentives to reconsider the conventional approach of distributing research funds via grant peer review.”

• Some references should be replaced by more recent ones.

Thank you, in addition to some outdated references, we noticed that some points needed more support. We have changed the references as follows: 

- Replaced Herbert et al 2013 (10.1136/bmjopen-2013-002800) by Schweiger 2023 (10.1371/journal.pone.0282320)

- Added Falk (2006) and Cialdini et al. (2006)

- Replaced Gillies 2014 (10.13130/2282-5398/3834) by De Peuter & Conix (10.1080/08989621.2021.1927727)

- Removed Jayasinghe (2003), as the same point was supported by other references already

- Replace Avin 2015 by Avin 2019 (10.1016/j.shpsa.2018.11.006) 

4) Reviewer 2:

This work is very interesting and presents a high degree of subjectivity that I believe has not been fully resolved.

Thank you for your comments, we are glad to hear you found the paper interesting. We’ve tried to improve the paper along the lines you suggested, and have rephrased some passages to avoid being overly subjective. We respond point-by-point here:

• The response rate was too low to resolve so many variables.

We are not sure we fully understand what the reviewer means by ‘resolving many variables’, and how this relates to the response rate (which we cannot estimate precisely due to the newsletter, cf. response to the editor’s comments). Each of the statistical models only seeks to estimate one particular variable. Given the causal assumptions we make (as specified in the DAGs), omitting variables from the models would bias our estimates, regardless of response rate or sample size. We do acknowledge that our sample size is limited, however this simply results in wider uncertainty intervals, which we think we communicate clearly in the relevant tables in the paper and supplementary materials. 

• Another situation that was not well resolved was if the questionnaire was sent to a European agency, obviously most respondents would be from the European continent (77.08%). It would be important to know which countries are on each continent, as there are important cultural differences that could be discussed and highlighted.

We explicitly state in the text that the funding agency is Flemish, that because of that most respondents came from Europe, and that this is a reason to be cautious about drawing strong conclusions from our data. We agree that it would be interesting to know which countries respondents were from, but (as explained in the paper) we did not ask this for privacy reasons: the combination of gender, field, seniority and country would make it possible to identify respondents, and our university’s ethics regulations compels us to only collect private information like that if we have very good reasons for this. We deemed protecting privacy more important than collecting country-level data, and hence did not include this in the demographic questions.

• Another statement that stands out is the one described in line 63 of the introduction. The fact that a junior researcher writes a funding project is part of a learning process, the important thing is that it is accompanied, guided, approved and sent by a senior, so as it is, I do not see it as bad conduct.

Thank you for this comment and example. It is in line with the editor’s comment that many of the behaviours we discuss might not be considered QRPs by all researchers. We have included the example in the introduction to make the point that they are grey-zone cases, but that we will consider anything that goes directly against commonly accepted codes of conduct as a QRP in this study. As co-authorship without acknowledgement is definitely against such codes of conduct, we think it should be considered a QRP even if the unnamed junior researcher benefits from it and the senior researcher only has good intentions doing it. The section in the introduction now reads:

“One can argue that many QRPs in the context of funding are situated in a grey zone between acceptable and unacceptable. For example, involving junior researchers in the writing of funding proposals can be part of a valuable learning process. However, when the junior researcher is not given due credit, problems with accountability, honesty and respect as they are described in various prominent codes of conduct do arise [14,15]. In this study, we will consider all behaviours that directly go against generally accepted scientific codes of conduct to be QRPs, even if some of these are considered “normal misbehaviours” [16].”

---

## [Decision Letter · Decision Letter 1]

10 Oct 2023

Questionable research practices in competitive grant funding: a survey

PONE-D-23-16082R1

Dear Dr. Conix,

We’re pleased to inform you that your manuscript has been judged scientifically suitable for publication and will be formally accepted for publication once it meets all outstanding technical requirements.

Kind regards,

Sonia Vasconcelos, PhD

Academic Editor

PLOS ONE

Additional Editor Comments (optional):

Reviewers' comments:

Reviewer's Responses to Questions

**Comments to the Author**

1. If the authors have adequately addressed your comments raised in a previous round of review and you feel that this manuscript is now acceptable for publication, you may indicate that here to bypass the “Comments to the Author” section, enter your conflict of interest statement in the “Confidential to Editor” section, and submit your "Accept" recommendation.

Reviewer #1: All comments have been addressed

Reviewer #2: All comments have been addressed

2. Is the manuscript technically sound, and do the data support the conclusions?

Reviewer #1: Yes

Reviewer #2: Yes

3. Has the statistical analysis been performed appropriately and rigorously? 

Reviewer #1: Yes

Reviewer #2: Yes

4. Have the authors made all data underlying the findings in their manuscript fully available?

Reviewer #1: Yes

Reviewer #2: Yes

5. Is the manuscript presented in an intelligible fashion and written in standard English?

Reviewer #1: Yes

Reviewer #2: Yes

6. Review Comments to the Author

Reviewer #1: The authors made significant efforts to revise their manuscript according to this reviewer's comments. Thank you.

Reviewer #2: The authors made the necessary clarifications for publication, answering the questions and making the necessary modifications

7. PLOS authors have the option to publish the peer review history of their article (what does this mean?). If published, this will include your full peer review and any attached files.

Reviewer #1: No

Reviewer #2: **Yes: **Sigmar de Mello Rode

---

## [Editor Report · Acceptance letter]

26 Oct 2023

PONE-D-23-16082R1 

Questionable research practices in competitive grant funding: a survey 

Dear Dr. Conix:

I'm pleased to inform you that your manuscript has been deemed suitable for publication in PLOS ONE. Congratulations! Your manuscript is now with our production department. 

Kind regards, 

on behalf of

Dr. Sonia Vasconcelos 

Academic Editor

PLOS ONE